# Effects of Probiotic-Fermented Corn Wet Distillers Grains on the Growth Performance, Carcass Characteristics, and Heavy Metal Residue Levels of Finishing Pigs

**DOI:** 10.3390/biology14081021

**Published:** 2025-08-08

**Authors:** Wang Liao, Xudong Wu, Zaigui Wang, Shuhao Fan

**Affiliations:** 1College of Animal Science and Technology, Anhui Agricultural University, No. 130 Changjiang West Road, Hefei 230036, China; wendy716@126.com; 2Anhui Provincial Key Laboratory of Livestock and Poultry Product Safety, Institute of Animal Husbandry and Veterinary Medicine, Anhui Academy of Agricultural Sciences, No. 40 Nongke South Road, Hefei 230031, China; white_wxd@163.com; 3School of Life Sciences, Anhui Agricultural University, No. 130 Changjiang West Road, Hefei 230036, China; wangzaigui2013@163.com

**Keywords:** fermented corn wet distillers grains, growth performance, carcass characteristics, heavy metal residue, pigs

## Abstract

Amid rapid agricultural development in China, there is an increasing focus on the utilization of alternative feed ingredients to promote sustainability and mitigate environmental pollutants in livestock production. Corn distillers’ grains, a byproduct of ethanol production, are often underutilized or discarded, thereby contributing to environmental waste. This study evaluated the potential of probiotic-fermented corn distillers grains as a sustainable feed resource in finishing pigs. Three *Bacillus subtilis* strains were used to ferment these byproducts, yielding three distinct fermented feed mixes. The results indicated that the *Bacillus subtilis*-fermented diets increased growth performance and improved the meat quality of pigs, and reduced the accumulation of harmful heavy metals in pig tissues. These findings offer valuable insights into the feasibility of using probiotic-fermented byproducts in livestock nutrition, contributing to more sustainable and safer feeding practices in swine production.

## 1. Introduction

With the rapid advances in China’s agricultural industry in recent years, interest in alternative feed ingredients for livestock has increased, driven by both the need for sustainable agricultural practices and the increasing pressure to reduce environmental pollutants in animal production [1]. Corn wet distillers grains (CWDGs), a by-product of the ethanol production process that consists of residual solids and soluble components that remain after alcohol has been extracted from corn, represent a widely available, economical, and resource-efficient feed option in China [2,3]. CWDGs typically contain 28–35% crude protein, 8–12% crude fat, and 30–40% neutral detergent fiber, along with essential amino acids such as lysine (1.2–1.8%) and methionine (0.5–0.7%), making them excellent alternative resources for animal feed [4,5]. However, the direct use of CWDGs as animal feed is limited by antinutritional factors such as mannan, arabinoxylan, and glucan, which reduce digestion efficiency and may cause stress in animals [6]. The increasing use of fermentation technology has led to the utilization of distillers grain resources, which could be an effective way to reduce natural antinutritional factors and improve palatability and utilization efficiency [7], making distillers grains attractive options for livestock nutrition.

Probiotics are used to ferment raw materials containing large amounts of nutrients under appropriate conditions to produce fermented feed. Among probiotics, *Bacillus subtilis* has been widely adopted in feed fermentation because of its robust enzymatic activity, thermostability, and ability to degrade complex polysaccharides such as arabinoxylan and β-glucan, which are the predominant antinutritional factors in CWDGs [8]. Furthermore, *Bacillus subtilis* is an organism generally recognized as safe (GRAS) by regulatory agencies, making it suitable for livestock applications [9]. Its ability to survive harsh gastrointestinal conditions and competitively exclude pathogenic bacteria in the gut further enhances its utility in improving animal health and nutrient absorption [10]. Microbial fermentation of distillers grains has been widely documented in the livestock industry, including applications in cattle, poultry, and swine, with reported improvements in production performance and meat quality [11,12,13]. For instance, studies have shown that feeding cattle diets containing fermented distillers grains can boost growth and intestinal immunity by modulating lipid metabolism [11]. Similarly, fermented distillers grains have been linked to enhanced production metrics and egg quality in laying hens [12], while their use in piglet diets has improved digestive efficiency and growth rates during extended nursery periods [13]. The fermentation of sorghum distillers’ grains has demonstrated benefits for carcass traits and meat quality in grower–finisher pigs [14]. Specifically, *Bacillus subtilis*-fermented feeds have demonstrated increased nutrient bioavailability in swine, which is attributed to the degradation of lignocellulosic structures and the release of bioactive peptides during fermentation [8]. In light of this possibility, fermented corn wet distillers grains (FCWDGs) could be considered a potential feedstuff for swine production.

While FCWDGs have shown promise in improving growth performance and meat quality in cattle, broilers, and pigs [11,12,13,14], the specific effects of *Bacillus subtilis*-fermented CWDGs on heavy metal accumulation in pig tissues—a critical concern for food safety—remain poorly understood. Arsenic (As), lead (Pb), cadmium (Cd), and copper (Cu) are particularly relevant due to their widespread environmental contamination, strict regulatory limits in meat products, and tendency to accumulate in agricultural byproducts such as corn distillers’ grains. These metals pose significant risks to both animal health and human consumption, as their residues in pork can compromise food safety and marketability.

This study addresses a significant gap in the literature by investigating the dual benefits of FCWDGs: enhancing growth and carcass traits while reducing heavy-metal residues in pig tissues. The novelty of this research lies in its focus on the combined effects of probiotic fermentation on both growth and tissue safety parameters in finishing pigs, using three distinct *Bacillus subtilis* strains to optimize fermentation outcomes. The findings are expected to provide valuable insights into the feasibility of using fermented byproducts in livestock nutrition and contribute to the development of more sustainable and safer feeding strategies for swine.

## 2. Materials and Methods

All procedures of this study were performed in accordance with the Chinese guidelines for animal welfare and were approved by the Committee for the Care and Use of Experimental Animals at Anhui Agricultural University (permit No. AHAU20101025).

### 2.1. Processing of Fermented Alcohol Byproducts for Incorporation in Animal Diets

CWDGs were obtained from the Anhui Academy of Agricultural Sciences (Hefei, Anhui, China) and contained 32.5% crude protein, 10.8% crude fat, and 35.2% neutral detergent fiber on a dry matter basis. The material was stored at −20 °C to prevent microbial spoilage until fermentation. The three strains of *Bacillus subtilis* used in this study were obtained from the China General Microbiological Culture Collection Center (CGMCC), China Center for Type Culture Collection (CCTCC), and China Center of Industrial Culture Collection (CICC). The strain CGMCC21218 is known for its high amylase activity, which can break down starch into simple sugars. The strain CCTCC2022073 possesses strong cellulase activity, assisting in cellulose degradation. The strain CICC10275 is characterized by its production of lactic acid and other organic acids, which can improve the palatability and nutrient bioavailability of the feed [15,16]. Strains were inoculated into NA liquid media (Hope Bio-Technology Co., Qingdao, China; Cat. No. HB0387), cultured at 37 °C and a rotational speed of 100 r/min, and oscillated for 48 h. Then, these cultured bacterial liquids were inoculated into NA liquid media, according to an inoculum amount of 2%, cultured at 37 °C and a rotational speed of 100 r/min, and oscillated for 24 h. When the number of viable bacteria reached 108/mL, the seed liquids of the three strains were obtained. CWDGs was used as the fermentation substrate, and strain CGMCC21218, CCTCC2022073, and CICC10275 seed liquors were added to the substrate at 12% of the total mass of the fermentation substrate and mixed evenly. The initial moisture content of the fermentation substrate was adjusted to 45%. After aerobic fermentation for 60 h at a fermentation temperature of 34 °C, an air-drying group was used to obtain the FCWDGs, which were denoted as FCWDGs-1, FCWDGs-2, and FCWDGs-3.

### 2.2. Feed and Animals

A total of 128 female Anqing six white pigs (54.64 ± 2.22 kg, about 130 days of age) were obtained from Anhui Huating Lake Green Food Co., Ltd., Anqing, China, and were randomly assigned to groups with four different experimental diets, each including 4 replicate pens with 8 pigs per pen. Subsequent to a preliminary 5-day period, the formal experiment period extends for a duration of 60 days. Pigs were provided free access to water under identical housing conditions throughout the experiment. Pigs fed a basal diet without FCWDGs supplementation were used as the control group, and those fed a basal diet supplemented with 6% FCWDGs-1 (T1), FCWDGs-2 (T2) or FCWDGs-3 (T3) comprised the experimental groups (Table 1).

### 2.3. Growth Performance

Pigs were weighed individually after 12 h of feed withdrawal at the beginning and end of the experiment, and the average daily gain (ADG) was calculated. The average daily feed intake (ADFI) was recorded on a pen basis every day. The feed-to-gain (F/G) ratio was calculated from the ADG and ADFI, where the F/G ratio = ADFI/ADG.

### 2.4. Measurements of Serum Hormone Levels

After 12 h of feed withdrawal, sixteen pigs from each group (four per replicate) were selected for blood sampling and for the determination of malonaldehyde (MDA), total antioxidative capacity (T-AOC), superoxide dismutase (SOD), glutathione peroxidase (GSH-px), cholecystokinin (CKK), secretin (SCT), and motilin (MTL) levels. Blood samples were collected from the precaval vein using heparinized tubes (2 mL) and immediately placed on ice for transport to the laboratory. Plasma was separated by centrifugation at 4 °C for 10 min (3500× *g*) and stored at −20° C until analysis. Concentrations of the target parameters were quantified using commercial analytical kits (Sigma, Thermo Fisher Scientific, Shanghai, China) and an autoanalyzer (Hitachi Ltd., Tokyo, Japan).

### 2.5. Carcass Traits

When pigs reached a body weight of 100 ± 5 kg at the end of the experiment, sixteen pigs from each group (four per replicate) were transported to a local abattoir near the experimental station. After 12 h of feed withdrawal, the pigs were stunned using electrical tongs (300 volts for 3 s) and then exsanguinated while suspended. The carcasses were subsequently placed in a dehairing unit set to 62 °C for 5 min, with residual hair removed using a knife and flame. Following evisceration, the carcasses were split and stored in a chiller at 5 °C for 12 h. The dressing percentage (DP) was calculated as the ratio of cold carcass weight to live weight after fasting. The backfat thickness (BF) and loin muscle depth (LMD) of the pigs were measured using a RENCO Series 12 ultrasound instrument (Beijing Tenovo Food Co., Ltd., Beijing, China) between the 10th and 11th ribs of the pigs. The phenotypic values of the lean meat percentage (LMP) were calculated on the basis of the BF and LMD as follows [17]:LMP (%) = 61.21920 − 0.77665 × BF + 0.15239 × LMD

### 2.6. Determination of Heavy-Metal Residues

Kidney, liver, and muscle samples were collected from the slaughtered pigs described above and then stored at −80 °C until chemical analysis. The samples were homogenized separately, and 10 g of freshly homogenized samples was weighed into porcelain dishes and evaporated to dryness in an oven at 105 °C. The ground samples were transferred to a porcelain basin and placed into a muffle furnace, and the temperature was increased gradually until a temperature of 550 °C was reached. The samples were digested with a triacid mixture (HNO_3_:HCO_4_:H_2_SO_4_) at a ratio of 10:4:1 at a rate of 10 mL per 10 g of sample and were placed on a hot plate at 100 °C. Digestion was allowed to continue until the liquor became clear. All the digested liquor was filtered through filter paper and diluted with distilled water. The determination of the heavy metals was performed directly on each final solution using a Buck Scientific 230 A atomic absorption spectrophotometer (Buck Scientific Instruments, LLC., Ansonia, CT, USA), following established protocols [18]. The concentrations of the heavy metals arsenic (As), lead (Pb), cadmium (Cd), and copper (Cu) were quantified from the calibration curves of the standards.

### 2.7. Statistical Analysis

The data were subjected to analysis of variance (ANOVA) using the PROC GLM procedure in SAS statistics software version 9.3 (SAS Institute Inc., Cary, NC, USA). All the data were tested for a normal distribution using the Kolmogorov–Smirnov test. Each replicate represented an experimental unit for data analysis. The significance of the differences between group means was tested using one-way ANOVA with Duncan’s multiple comparison; *p* < 0.05 was considered significant. The data are expressed as the means ± SDs.

## 3. Results

### 3.1. Productive Performance

The growth performance of the finishing pigs in the different groups is shown in Table 2. The initial weights were similar among all the groups. Compared with that of the control, the final weight was significantly greater in both T1 and T3 (*p* < 0.05), resulting in a notable increase in the ADG (*p* < 0.05). No significant differences in the final weight or ADG were noted between the control and T2 treatments (*p* > 0.05). Compared with that of the control, the ADFI was significantly greater in T3 (*p* < 0.05) and significantly lower in T1 (*p* < 0.05). No significant difference in the ADFI was observed between the control and T2 groups (*p* > 0.05). Significant differences in the F/G ratios were detected among the four groups (*p* < 0.05), with the control group exhibiting the highest value, followed by T2, T3, and T1, in ascending order.

### 3.2. Slaughter Performance

The carcass traits of the finishing pigs in the different groups are shown in Table 3. Compared with that of the control, the percentage of LMD was significantly greater in pigs from the T1, T2, and T3 groups (*p* < 0.05). The percentage of LMP was significantly greater in pigs from the T1 group (*p* < 0.05) and slightly greater in pigs from the T2 and T3 groups (*p* > 0.05) compared with the control. No significant differences in DP% or BF were detected among the four groups (*p* > 0.05).

### 3.3. Serum Biochemical Parameters

The serum parameters of the finishing pigs in the different groups are shown in Table 4. Compared with those in the control, the serum MDA levels were significantly lower in the T1 group (*p* < 0.05) and slightly lower in the T2 and T3 groups (*p* > 0.05), whereas the serum T-AOC levels were significantly greater in the T1 group (*p* < 0.05) and slightly greater in the T2 and T3 groups (*p* > 0.05). The serum SOD and GSH-PX levels in the T1, T2, and T3 groups were significantly greater than those in the control group (*p* < 0.05), with the highest levels observed in the T1 group. Although the serum CKK, SCT, and MTL levels were greater in the T1, T2, and T3 groups than in the control group, no significant differences were noted among the four groups (*p* > 0.05).

### 3.4. Heavy Metal Residue

The heavy-metal residues of the finishing pigs in the different groups are shown in Table 5. Compared with those in the control group, the levels of arsenic (As) in the muscle, liver, and kidney tissues decreased in the T1, T2, and T3 groups. However, the T1 group presented a notably lower level of As in the kidney than the other groups (*p* < 0.05). Compared with those in the control group, the cadmium (Cd) levels in muscle, liver and kidney tissues were slightly lower in the T2 and T3 groups (*p* > 0.05), whereas a significant reduction was noted in the T1 group (*p* < 0.05). Furthermore, the Cd level in the T1 group was significantly lower than that in the T2 and T3 groups (*p* < 0.05). Compared with those in the control group, the levels of Pb in the muscle, liver and kidney were markedly lower in the T1, T2, and T3 groups (*p* < 0.05), with the T1 group exhibiting significantly lower levels than the T2 and T3 groups (*p* < 0.05). The copper (Cu) content in muscle was significantly lower in the T2 and T3 groups (*p* < 0.05) than in the control group, whereas the Cu content in the liver and kidney was slightly lower (*p* > 0.05). The Cu content in the muscle, liver and kidney was significantly lower in the T1 group than in the control group (*p* < 0.05) and significantly lower than that in the T2 and T3 groups (*p* < 0.05).

## 4. Discussion

The results revealed that dietary supplementation with FCWDGs, particularly in the T1 and T3 groups, significantly increased the average daily gain (ADG) and final body weight compared with those of the control group, demonstrating that microbial fermentation can improve feed intake by reducing antinutritional factors [19]. This improvement aligns with the distinct enzymatic profiles of the *Bacillus subtilis* strains used in T1 and T3. The high amylase activity of strain CGMCC21218 used in the T1 group may contribute to facilitating the release of nutrients embedded within CWDGss, and strain CICC10275 used in the T3 group has the potential to increase nutrient absorption efficiency by modulating the intestinal flora of pigs through the production of lactic acid [15,16]. However, T2 had no significant effect on growth performance, suggesting that not all probiotic strains may confer equivalent benefits. This variation may be due to differences in the ability of probiotic strains to ferment CWDGs and produce beneficial metabolites such as organic acids and enzymes, which are known to aid in digestion and nutrient absorption. Similar results have been reported in other studies investigating the effects of fermented distillers grains on livestock. For example, feeding fermented sorghum distillers grains improved the growth performance of pigs [14], which was attributed to enhanced nutrient digestibility and gut health benefits. The results from the present study revealed that the F/G ratio was significantly lower in the pigs that received FCWDGs than in the pigs fed a basal diet. The lowest value was observed in the T1 group, indicating superior feed conversion efficiency in the T1 group compared with that of the T2 and T3 groups. Similarly, Tang et al. [20] reported that fermented complete feed improved growth performance by reducing the feed-to-gain ratio. These findings indicate the potential use of *Bacillus subtilis* to increase productive performance in finishing pigs.

The present study revealed that FCWDGs supplementation can positively influence the lean meat percentage (LMP) and loin muscle depth (LMD) in finishing pigs, especially in the T1 group. LMP and LMD are key indicators of meat quality for swine because they reflect the proportion of lean muscle tissue relative to fat, which directly affects the yield of high-quality meat products. This is likely because FCWDGs can increase the concentration of bioactive compounds, including volatile fatty acids, which may have an impact on muscle development [21]. In addition, the enhanced amino acid profile of the fermented feed could significantly contribute to skeletal muscle accretion in finishing pigs by promoting protein synthesis and reducing muscle degradation in the longissimus thoracis. [22]. This aligns with the higher LMD values in T1 (47.21 mm) and T3 (49.06 mm) than in the control (44.78 mm), which may reflect increased muscle protein synthesis driven by bioavailable amino acids. Similar improvements in carcass traits have been reported in other studies evaluating the impact of fermented feed on pigs. Qiu et al. [23] demonstrated that a fermented diet significantly enhanced loin eye area and lean meat percentage while reducing backfat thickness and improving meat quality in late-finishing pigs compared to control-fed pigs. Furthermore, Ahmed et al. [24] reported that fermented herb supplementation decreased backfat thickness and meat crude fat content, resulting in higher lean meat percentages in grower–finisher pigs.

Oxidative stress is a known concern in livestock production, as it can negatively affect growth, immune function, and meat quality [25]. The significant reduction in MDA levels as well as increases in T-AOC levels and SOD and GSH-Px activities in the FCWDG groups, particularly in T1, indicated an increase in the oxidative status of the pigs. This finding was directly correlated with the strain’s ability to degrade reactive oxygen species (ROS). Strain CGMCC21218 (T1) likely produced antioxidants during fermentation, which were absorbed systemically, as evidenced by the highest SOD (64.25 U/mL) and GSH-Px (1710.52 U/mL) levels occurring in the T1 group. The observed improvements in antioxidant parameters suggest that probiotic fermentation of CWDGs may help alleviate oxidative stress, potentially through the production of bioactive compounds by probiotics, which can enhance pig immune responses and general health [26]. This result aligns with the findings of Wang et al. [27], who reported that supplementation with *Lactobacillus fermentum* increased antioxidant enzyme activity and improved meat quality in grower–finisher pigs. Interestingly, although the serum levels of cholecystokinin (CKK), secretin (SCT), and motilin (MTL) were greater in the treatment groups than in the control groups, no significant differences were detected among the groups. These hormones play a role in digestive function and appetite regulation, which could explain the improved feed intake and growth performance of the treatment groups. However, the lack of statistically significant differences in CKK, SCT, and MTL levels suggested that probiotic fermentation may not have a substantial effect on gastrointestinal hormone regulation in pigs or that the effects might be too subtle to detect with the sampling methodology employed.

The study also highlighted the ability of probiotic-fermented CWDGs to reduce heavy-metal residues, particularly those of arsenic (As), cadmium (Cd), lead (Pb), and copper (Cu), in muscle, liver, and kidney tissues. The T1 group, in particular, presented the lowest concentrations of these heavy metals, suggesting superior degradation ability. This finding is particularly important given the growing concerns over heavy metal contamination in livestock products and its potential impact on human health. Probiotic microorganisms can interact with heavy metals in the gastrointestinal tract, binding to them and reducing their absorption into tissues and the bloodstream [28]. The observed reduction in heavy-metal concentrations in pig samples fed FCWDGs may be attributed to several mechanisms. First, the cell walls of *Bacillus subtilis* and other microbial cells can bind to heavy metals, reducing their bioavailability [9]. Second, fermentation may produce chelating agents that can complex with heavy metals, preventing their absorption into tissues [8]. Finally, improved gut health and increased nutrient bioavailability in pigs fed FCWDGs may be conducive to reducing the absorption of heavy metals from the gastrointestinal tract. Despite current findings, more research is essential to clarify the mechanisms by which FCWDGs influence heavy metal removal in fecal samples. Previous studies have shown that probiotic microorganisms, including strains of *Bacillus* species, exhibit heavy-metal binding capacity, high heavy metal tolerance, and other characteristics that are beneficial for the host [29]. The specific probiotic strains used in this study may possess unique properties that facilitate the detoxification of these metals. Furthermore, increased antioxidant activity was observed in the probiotic groups, which could help mitigate the oxidative damage caused by the accumulation of heavy metals, which are known to induce oxidative stress in animals.

## 5. Conclusions

This study demonstrates that *Bacillus subtilis*-fermented corn wet distillers grains (FCWDGs) serve as a valuable feed ingredient for improving growth performance, carcass characteristics, and meat quality in finishing pigs. FCWDGs supplementation significantly boosted growth metrics, enhanced antioxidant capacity, and reduced heavy-metal residues in tissues, with FCWDGs-1 (CGMCC21218) showing the most pronounced benefits. These findings underscore the potential of probiotic-fermented byproducts to enhance swine production sustainability and food safety. The results provide critical insights into the feasibility of integrating FCWDGs into livestock diets, advancing sustainable and safe feeding practices in the swine industry.

## 6. Future Perspectives

While this study highlights the potential of *Bacillus subtilis*-fermented corn wet distillers grains (FCWDGs) to enhance swine productivity and safety, limitations such as strain-specific variability, long-term efficacy, and environmental scalability require further exploration. A critical limitation of this study is the absence of an unfermented CWDGs control group, which restricts the ability to distinguish the effects of fermentation from those of the raw material itself. Future research should validate its applicability across livestock species (e.g., poultry, ruminants) and investigate synergistic probiotic combinations or enzyme-assisted fermentation to optimize nutrient utilization and gut health. Additionally, mechanistic studies on microbial interactions, heavy metal detoxification pathways, and economic feasibility are critical for practical adoption. These efforts could advance sustainable feed strategies, reduce environmental impacts, and strengthen food safety standards, offering broader implications for global livestock systems and policy development.

## Figures and Tables

**Table 1 biology-14-01021-t001:** Ingredients and nutrient levels in the experimental diets.

Item	Group	
	Control	Treatments
Corn	62.00	59.95
Soybean meal	5.00	4.00
Wheat	19.00	18.50
Sunflower meal	7.00	5.20
Palm kernel meal	4.50	3.85
FCWDGs	0	6.00
Premix ^1^	2.50	2.50
Total	100.00	100.00
Nutrient level (as-fed basis)		
Crude protein	12.82	12.81
Digestible energy (MJ/kg)	12.97	12.97
Lysine	0.48	0.46
Methionine	0.23	0.24
Threonine	0.46	0.45
Methionine + Cystine	0.47	0.50
Calcium	0.56	0.57
Phosphorus	0.18	0.19

^1^ Provided the following per kilogram of feed: VA, 5000 IU; VD, 3000 IU; VE, 12 IU; VK3, 1.2 mg; VB1, 1 mg; VB2, 4 mg; VB6, 2 mg; VB12, 0.02 mg; niacin, 20 mg; folic acid, 1 mg; pantothenic acid, 9.2 mg; choline, 400 mg; Cu, 100 mg; Fe, 250 mg; Zn, 120 mg; Mn, 45 mg; I, 0.7 mg.

**Table 2 biology-14-01021-t002:** Effects of dietary fermented corn wet distillers grains (FCWDGss) on the growth performance of finishing pigs.

Items	Groups ^1^			
	Control	T1	T2	T3
Initial weight (kg)	54.60 ± 2.51	54.54 ± 2.72	54.61 ± 2.33	54.58 ± 2.49
Final weight (kg)	95.71 ± 3.87 ^b^	101.23 ± 3.83 ^a^	97.56 ± 3.77 ^b^	100.67 ± 3.96 ^a^
Average daily gain (kg/day)	0.69 ± 0.02 ^b^	0.78 ± 0.03 ^a^	0.72 ± 0.03 ^b^	0.77 ± 0.03 ^a^
Average daily feed intake (kg/day)	2.65 ± 0.05 ^b^	2.58 ± 0.04 ^c^	2.62 ± 0.04 ^b,c^	2.72 ± 0.05 ^a^
F/G ratio ^2^	3.84 ± 0.14 ^a^	3.31 ± 0.08 ^d^	3.63 ± 0.10 ^b^	3.53 ± 0.12 ^c^

^a,b,c,d^ Means with different superscripts in the same row are different at *p* < 0.05. ^1^ Control—0% FCWDGs; T1—6% FCWDGs-1; T2—6% FCWDGs-2; T3—6% FCWDGs-3. ^2^ F/G ratio = ADFI/ADG.

**Table 3 biology-14-01021-t003:** Effects of dietary fermented corn wet distillers grains (FCWDGs) on the carcass traits of finishing pigs.

Group ^1^	Carcass Trait ^2^			
	DP (%)	BF (mm)	LMD (mm)	LMP (%)
Control	73.78 ± 1.77	33.07 ± 1.12	44.78 ± 2.29 ^b^	43.02 ± 1.10 ^b^
T1	74.91 ± 2.31	31.31 ± 1.08	47.21 ± 2.37 ^a^	44.57 ± 1.17 ^a^
T2	75.14 ± 2.19	32.55 ± 1.19	48.33 ± 2.64 ^a^	44.13 ± 1.23 ^a,b^
T3	75.02 ± 2.08	32.14 ± 1.14	49.06 ± 2.71 ^a^	44.09 ± 1.19 ^a,b^

^a,b^ Means with different superscripts in the same column are different at *p* < 0.05. ^1^ Control—0% FCWDGs; T1—6% FCWDGs-1; T2—6% FCWDGs-2; T3—6% FCWDGs-3. ^2^ DP—dressing percentage; BF—backfat thickness; LMD—loin muscle depth; LMP—lean meat percentage.

**Table 4 biology-14-01021-t004:** Effects of dietary fermented corn wet distillers grains (FCWDGs) on the serum parameters of finishing pigs.

Item ^1^	Group ^2^			
	Control	T1	T2	T3
MDA (nmol/mL)	2.96 ± 0.37 ^a^	2.31 ± 0.32 ^b^	2.70 ± 0.29 ^a^	2.84 ± 0.23 ^a^
T-AOC (μmol/L)	152.81 ± 11.22 ^b^	216.25 ± 17.80 ^a^	159.32 ± 12.95 ^b^	162.47 ± 11.57 ^b^
SOD (U/mL)	46.83 ± 3.61 ^c^	64.25 ± 4.37 ^a^	56.68 ± 3.43 ^b^	57.78 ± 4.05 ^b^
GSH-px (U/mL)	1229.81 ± 119.81 ^c^	1710.52 ± 129.40 ^a^	1472.55 ± 125.74 ^b^	1509.27 ± 130.61 ^b^
CKK (pg/mL)	0.38 ± 0.03	0.38 ± 0.04	0.38 ± 0.03	0.38 ± 0.03
SCT (pg/mL)	0.37 ± 0.03	0.39 ± 0.03	0.38 ± 0.03	0.38 ± 0.03
MTL (pg/mL)	0.32 ± 0.02	0.32 ± 0.02	0.32 ± 0.02	0.32 ± 0.03

^a,b,c^ Means with different superscripts in the same row are different at *p* < 0.05. ^1^ MDA—malonaldehyde; T-AOC—total antioxidative capacity; SOD—superoxide dismutase; GSH-px—glutathione peroxidase; CKK—cholecystokinin; SCT—secretin; MTL—motilin. ^2^ Control—0% FCWDGs; T1—6% FCWDGs-1; T2—6% FCWDGs-2; T3—6% FCWDGs-3.

**Table 5 biology-14-01021-t005:** Effects of dietary fermented corn wet distillers’ grain (FCWDG) on heavy-metal residues in finishing pigs.

Organs	Item ^1^	Group ^2^			
		Control	T1	T2	T3
Muscle	As	0.07 ± 0.02	0.06 ± 0.01	0.06 ± 0.02	0.06 ± 0.02
	Pb	507.34 ± 36.41 ^a^	356.45 ± 22.35 ^c^	446.67 ± 31.50 ^b^	455.28 ± 29.87 ^b^
	Cd	308.13 ± 26.86 ^a^	207.61 ± 15.43 ^b^	277.43 ± 19.16 ^a^	271.48 ± 21.52 ^a^
	Cu	2178.62 ± 156.37 ^a^	1424.57 ± 112.10 ^c^	1806.45 ± 128.95 ^b^	1906.55 ± 125.41 ^b^
Liver	As	0.46 ± 0.04	0.42 ± 0.03	0.43 ± 0.03	0.43 ± 0.03
	Pb	687.56 ± 52.43 ^a^	471.32 ± 29.78 ^c^	554.67 ± 34.60 ^b^	567.36 ± 37.22 ^b^
	Cd	777.56 ± 56.61 ^a^	587.23 ± 38.94 ^b^	774.17 ± 60.11 ^a^	769.45 ± 55.19 ^a^
	Cu	2037.43 ± 136.35 ^a^	1636.67 ± 98.52 ^b^	1904.76 ± 121.56 ^a^	1932.35 ± 125.68 ^a^
Kidney	As	0.75 ± 0.09 ^a^	0.40 ± 0.02 ^b^	0.73 ± 0.05 ^a^	0.70 ± 0.03 ^a^
	Pb	514.23 ± 26.56 ^a^	353.23 ± 16.15 ^c^	487.41 ± 21.87 ^b^	479.43 ± 22.34 ^b^
	Cd	2534.16 ± 173.62 ^a^	1556.35 ± 122.40 ^b^	2234.67 ± 169.44 ^a^	2389.56 ± 173.31 ^a^
	Cu	2356.07 ± 185.32 ^a^	1334.56 ± 112.97 ^b^	2113.12 ± 179.35 ^a^	2235.56 ± 172.51 ^a^

^a,b,c^ Means with different superscripts in the same row are different at *p* < 0.05. ^1^ As—arsenic; Pb—lead; Cd—cadmium; Cu—copper. ^2^ Control—0% FCWDGs; T1—6% FCWDGs-1; T2—6% FCWDGs-2; T3—6% FCWDGs-3.

## Data Availability

The data presented in this study are available on request from the corresponding author, due to confidentiality agreements.

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
