# Peer review of "Effects of Probiotic-Fermented Corn Wet Distillers Grains on the Growth Performance, Carcass Characteristics, and Heavy Metal Residue Levels of Finishing Pigs"

_biology, 2025, doi:10.3390/biology14081021_

Round 1
Reviewer 1 Report
Comments and Suggestions for Authors
This study addresses a relevant topic and provides clear evidence that Bacillus-fermented wet distillers' grains improve the growth and carcass traits of finishing pigs, while reducing heavy metal residues. The experimental design is adequate, the data are analysed correctly, and the discussion is largely balanced. However, some minor issues remain:
- Please indicate the experimental period in the abstract.
- The experiment did not include an unfermented CWDGS control group, which makes it impossible to distinguish between the effects of fermentation and the raw material itself. It is therefore recommended that a 6% unfermented CWDGS group is added, or that this limitation is at least explained in the discussion.
- Line 129: Please provide the gender or the male-to-female ratio for the pigs.
- Lines 105–127: The fermentation conditions (aerobic/anaerobic) were not specified. Bacillus subtilis mostly ferments through aerobic fermentation.
- Lines 328–348: The lack of heavy metal excretion in fecal samples cannot directly prove 'reduced heavy metal absorption'.
- All statistical marks in the tables need to be checked again to ensure consistency with the results.
Reviewer 2 Report
Comments and Suggestions for Authors
Dear authors,
the study appears to be novel in several important ways, and as there’s increasing interest in environmental sustainability in agriculture worldwide, and applying this approach in the context of China, where ethanol production and waste management are big issues, makes the research highly relevant and timely.
Please rewrite the abstract to highlight the important findings of the study, and same counts for the conclusions.
In the introduction highlight please the need of this this study and why is novel.
Add a new section of future perspectives, it will be a strong addition to your study, and include here such as limitations that future studies should address, opportunities for expanding the research or implications for industry or policy. The approach could be tested on other livestock species (e.g., poultry, ruminants) to evaluate the generalizability of the probiotic-fermented feed concept? The combinations of different probiotic strains or co-fermentation with enzymes may further enhance feed efficiency and animal health outcomes?
Please confirm that pigs were randomly allocated to pens and treatments to avoid bias.
Explain why the focusing on arsenic (As), lead (Pb), cadmium (Cd), and copper (Cu) is scientifically justified and strategically relevant for several reasons. Many countries, including China, have strict residue limits for these metals in meat products, and these metals can accumulate in agricultural byproducts, including corn distillers’ grains. Their levels directly affect the safety and marketability of pork products.
